# Vegetation Affecting Water Quality in Small Streams: Case Study in Hemiboreal Forests, Latvia

**DOI:** 10.3390/plants11101316

**Published:** 2022-05-16

**Authors:** Mārcis Saklaurs, Stefānija Dubra, Līga Liepa, Diāna Jansone, Āris Jansons

**Affiliations:** 1Latvian State Forest Research Institute Silava, Rigas Street 111, LV-2169 Salaspils, Latvia; marcis.saklaurs@silava.lv (M.S.); diana.jansone@silava.lv (D.J.); aris.jansons@silava.lv (Ā.J.); 2Forestry Faculty, Latvia University of Life Sciences and Technologies, Liela 2, LV-3001 Jelgava, Latvia; liga.liepa@llu.lv

**Keywords:** riparian forests, species composition, ecosystem service, forest management, small streams

## Abstract

Riparian forests are important ecosystems located along the margins of freshwaters. Riparian zones provide many ecosystem services, such as nutrient modification, erosion and temperature control, leading to improvements in water quality in adjacent water ecosystems. In many areas, riparian forest management is restricted to improve adjacent water quality. The potential influence of forest management on water quality of small streams was assessed by analysing species composition and structural diversity in riparian forests. We collected data in riparian forests along 15 streams in the eastern Baltics (Latvia) with different water quality classes. We used detrended correspondence analysis and indicator species’ analysis to determine relationships between woody plants and understory vegetation. We also used ADONIS and ANOSIM analysis to determine possible factors that affect species composition. Our results suggested that water quality is affected by ground vegetation, which in turn was altered by stand density and total yield. Site-specific decision making in management is required in riparian forests to ensure the required conditions in the streams, because species composition differs between sites, dominant tree species and stand parameters (density, total yield, stand age). Introduction of *Betula pubescens* Ehrh. in coniferous stands is favourable to ensure litter fall quality and provide shade for streams during summer.

## 1. Introduction

Riparian forests are diverse ecosystems located along the margins of freshwaters [1]. The transition zone between aquatic and terrestrial systems provides optimal environmental conditions for aquatic biota, especially rare species [2,3]. In addition, riparian forests provide many environmental services by performing as pollutant sinks, and ensuring stream water quality [4,5,6] and high structural diversity in riparian areas is beneficial for aquatic ecosystem functioning [7]. Several factors, including biophysical and geographical parameters such as spatial extent, soil dynamics, distributions and characteristics related to succession, determine diversity and species composition in riparian areas [8]. Besides preserving biodiversity in riparian forests, it is important to understand the interrelationship between forested areas and aquatic ecosystems. 

Nutrient leaching into streams is one of the main causes of species decline in freshwaters [9,10]. Nitrogen and phosphorus enrichment in streams cause eutrophication, thereby subtracting oxygen and light availability for umbrella species [11]. Organic matter flux into a stream causes water over-enrichment, impairs diversity of invertebrates and other aquatic organisms, changes microbial composition, and leads to organic pollution in the catchment area [12,13,14,15]. Small streams are an important ecosystem for umbrella species [16,17]. During the last decades there has been a decline in brown trout (*Salmo trutta fario* L.), pearl mussel (*Margaritifera margaritifera* L.) and thick-shelled river mussel (*Unio crassus Philipsson*) in Europe, including in Latvia [13,17,18]. Oxygen availability is crucial to the survival of these species. For instance, brown trout is sensitive to stream chemical and physical alterations—clean, cool and well-oxygenated water are the requirements for optimal growth conditions [16]. Freshwater pearl mussel has experienced an extreme decline in population in Europe [16,17]. Only a few locations of populations with optimal reproduction have been found [16]. Furthermore, pearl mussel populations are highly dependent on host fish populations to complete metamorphosis [19]. The viable host species for pearl mussel are sea trout (*Salmo trutta f. trutta*) and above-mentioned brown trout [20]. Previous studies have suggested that in riparian areas, aquatic systems are positively reflecting on biodiversity and species composition of ground cover vegetation and trees in adjacent forested area [13,21,22]. 

Forest understory vegetation and woody plants are involved in water quality provision via nutrient retention and sequestration of organic matter [23]. Nutrient uptake in riparian areas is based on water movement through the rooting zone of plants as well as on the microbial uptake and sorption [13,22]. The capacity to retain nutrients, organic matter and other elements depends on soil type and hydrology of the area, but species composition and plant characteristics play an important role in nutrient flux [21,24,25]. In riparian areas, nitrogen and phosphorus removal is more effective in soils with tree cover, representing plant communities with deep root system, in comparison to shrub or grass vegetation [23]. Furthermore, perennial plants can accumulate environmental pollutants over several years [5]. Many studies in riparian areas have been focused on the Salicaceae family because of its inherent characteristics such as wide-spreading root system [26] with high filtering capacity [27,28], efficient nutrient uptake from soil, and relatively fast growth [29]. The extensive root system also delays erosion processes in sloped areas [27]. Although Salicaceae species are suitable for phytoremediation purposes, these species are preferred by beavers [30]. Beavers’ impact on water quality differs between the type of ecosystem. Beavers create dams, thereby increasing the retention of organic matter and sediments and altering local conditions for aquatic habitat [31]. Damming activities by beavers cause tree to fall, which creates open areas in riparian forests; this causes changes in species distribution and composition of trees [32]. Beaver activity limits migration of umbrella species, aquatic invertebrates and salmonid fish, reduces streams’ self-cleaning capacity and degrades the overall ecosystem of a stream [33]. 

In addition, plants play an important role in riparian ecosystems as slope stability performers [29,34]. Vegetation’s impact on slope stability has mechanical and hydrological effects—it stabilizes soil trough the root system and reduces soil water content trough evapotranspiration and transpiration processes [13]. In comparison to herbaceous plants, shrubs and trees are more effective against erosion and landslides because of their wider and deeper root systems [34]. However, the mechanical stability on an ecosystem level is provided by root reinforcement because roots support above-ground biomass by anchoring themselves; this reduces the probability of shear failure [35,36]. In previous studies on riparian forests, much attention has been drawn to woody and herbaceous plants, but bryophytes in this context are poorly studied. In boreal zones, mosses are an abundant part of the understory layer and provide high biodiversity to the ecosystem [37,38]. Bryophytes are essential for forest ecosystems—they regulate above- and below-ground temperature as well as provide nitrogen fixation via cyanobacteria [39,40]. Bryophytes respond to the changes and conditions in the forest microclimate [41]. During the studies on riparian areas, it is important to incorporate each level of the ecosystem to understand the whole impact on streams. 

We assessed the linkage between vegetation of riparian zones, including trees, understory herbaceous plants and bryophytes, and differences in stream water quality. The aim of our study was to evaluate direct (stand parameters) and indirect (species composition) factors’ effect on water quality classes of small streams. 

## 2. Materials and Methods

### 2.1. Location of Sample Plots

The study was conducted in the hemiboreal zone in Latvia (55.40°–58.05° N and 20.58°–28.14° E). We selected 15 streams shorter than 100 km (Figure 1), which flowed through forest complexes (at least four km along the stream). Selected riparian forests were generally mixed forests, dominated by Norway spruce (*Picea abies* L.), Scots pine (*Pinus sylvestris* L.), grey alder (*Alnus incana* L.), birch (*Betula* spp.), black alder (*Alnus glutinosa* L.) and European aspen (*Populus tremula* L.). The dominant forest vegetation types were *Aegopodiosa, Hylocomiosa* and *Vacciniosa*, but almost all forest types were represented in sample plots. 

### 2.2. Field Survey

The field survey ran from July to August of 2014 (approximately 2 months of vegetation data collection). For each stream, we established three transects perpendicularly from the river bank interior with a distance 1 km from each other (due to the forest complexed being at least four km-long). We chose sample plots avoiding forest stands with any signs of management activities during the last decade. The distance between transects was 1 km along the stream. At each transect, we placed two sample plots with sizes of 20 × 20 m (400 m^2^) at distances of 10 and 60 m from the water edge (Appendix A). In each sample plot, relative projective coverage of moss, herb, shrub and tree layers was recorded by species [42]. The nomenclature for bryophytes followed Āboliņa and for vascular plants Gavrilova and Šulcs [43,44]. In each sample plot, diameter at breast height (DBH) and height was measured for all trees with DBH > 6 cm. 

### 2.3. Stream Parameters

We used water quality data provided by the Latvian environment geology and meteorology centre [45]. The quality of stream included following parameters: biological parameters (occurrence of macrozoobenthic and fish species) and physicochemical parameters such as biological oxygen demand, phosphorus and nitrogen concentrations [46]. Based on these parameters, all selected streams fell into one out of the following categories of water chemical purity—low, moderate and high quality of water [45]. For each sample plot, we calculated stream power index (SPI) based upon slope and contributing area to describe protentional erosion flow at specific sites [47]. SPI values indicate if area is potentially erosive or relatively flat, which influences the susceptibility to floods. SPI was determined using SPIN toolbox in ArcGIS environment [48]. 

### 2.4. Data Analysis

All data were analysed using R Studio under version 1.3.1093 [49]. To assess the main possible forest stand parameters (total yield, total density, dead wood) and SPI that may have impact on species composition in riparian forests, we used permutational multivariate analysis of variance using distance matrices (ADONIS), using package “vegan” [50]. In our analysis we included all bryophytes, understory vegetation, including herbaceous, shrub and woody plants under 1.50 m height, and tree species. To visualize species composition and determine the factors (total yield, total density, SPI) significantly affecting species composition, we used detrended correspondence analysis (DCA) under “vegan” package [50]. We divided vegetation species data into three categories—the first category (C1) consists of moss species, herbaceous plant species and shrub/tree layer species under 1.5 m height. The second category (C2) includes only all tree species with DBH > 6 cm. The third category (C3) consists of only moss species. Rare species were downscaled and axes were rescaled. To understand the factors affecting similarity or difference in species composition between all studied sites, we used analysis of similarity (ANOSIM). We performed Indicator Species Analysis using package “indiscspecies” [51] to determine which species associate with the three water quality classes for the chemical purity of streams. Species from all categories were included in the analysis. 

Additionally, we performed indicator species analysis for the chemical purity of streams based on vegetation data included in ADONIS analysis [45]. 

## 3. Results

Altogether, 265 species were detected and included in further analysis.

### 3.1. Species Composition

All moss, understory and woody species were included in the analysis. ADONIS analysis indicated a significant influence of total yield and stand total density (*p* < 0.05) on species composition in sample plots, but it is important to note that there is relatively small variation (17%) within factors in the analysis (Table 1). Deadwood and SPI did not have significant impacts on understory species composition in studied sample plots.

ANOSIM indicated that species composition lightly differs between studied sites (streams) (R^2^ = 0.52; *p* < 0.05). However, similar species composition was found between different distances from the stream margins and different water quality classes (Table 2). In the analysis, we included bryophytes and herbaceous plant species. 

We performed detrended correspondence analysis (DCA) for each of the three vegetation species categories C1, C2 and C3 (Figure 2 and Figure 3). The ordination of the combined data from C1 category (DCA_C1_) gave the following eigenvalues for the first and second axes: 0.62 and 0.35. DCA_C1_ shows the species composition among different sites with different forest types. Total yeld had a significant correlation (R = 0.44; *p* < 0.05) with Hylocomiosa and Vacciniosa forest vegetation species (*Calamagrostis arundinacea* L., *Agrostis stolonifera* L., Ribes spicatum E. Robson, *Huperzia selago* L., *Orthilia secunda* L., *Rubus saxatilis* L., *Mycelis muralis* L., *Melampyrum pratense* L., *Corylus avellana* L., *Trientalis europaea* L., Pleurozium schreberi, Sphagnum angustifolium and Sphagnum girgensohnii) (Figure 2). Total yield and SPI had a significant correlation (*p* < 0.05) with specific species (Figure 2), but the variation is low (R^2^ < 0.1). DCA_C1_ also represents the distribution of dominant forest vegetation types—Aegopodiosa, Hylocomiosa and Vacciniosa. Of all the vascular plants, 47 species had Ellenberg indicator values for nitrogen 7–10, indicating their ability to grow in fertile soils. Furthermore, species that prefer sites rich in nitrogen (for instance, *Aegopodium podagraria* L., *Urtica dioica* L., and *Stellaria media* L.) were found in fertile soils and in presence of *Alnus incana* L. and Padus avium Mill. Mosses that prefer fertile soils were Plagiomnium affine, Plagiomnium cuspidatum and Eurhynchium angustirete. Grass species that preferred oligotroph conditions were *Molinia caerulea* L., *Deschampsia flexuosa* L., *Calamagrostis arundinacea* L., and *Poa nemoralis* L., but mosses found in oligotroph soils were mostly Polytrichum commune, Polytrichum juniperinum, Ptilium crista-castrensis, *Dicranum majus*, Sphagnum angustifolium and Sphagnum girgensohnii. These species were found in the presence of *Picea abies* L. and *Betula pubescens* Ehrh.

The DCA of the combined data from C2 gave the following eigenvalues for the first and second axes: 0.62 and 0.27 (Figure 3). Total yield had a significant correlation (R = 0.44; *p* < 0.05) with following tree species: *Picea abies* L., *Pinus Sylvestris* L., *Ulmus laevis* Pall., *Betula pendula* Roth. And *Tilia cordata* Mill. (Figure 3). The DCA of the combined data from C3 gave the following eigenvalues for the first and second axes: 0.75 and 0.45 (Figure 4). The occurance of following moss species *Hylocomium splendens*, *Polytrichum commune*, *Rhytidiadelphus squarrosus* and *Rhytidiadelphus triquetrus* corelate with total yield (R = 0.44; *p* < 0.05). 

### 3.2. Bioindicator Analysis

The results show the plant species that are significantly associated with each of the water quality classes and each combination of the two classes (Appendix A). In total, 265 species were used for analysis, but 37 species were associated with a specific water quality group. Seven species were associated with only one group, but 13 species were associated with two classes. From the selected species 7 species were found only in riparian forests with streams of high water quality, 11 with streams with medium water quality, and 6 species were found in areas with streams with low water quality. 

Species that significantly belong to streams with high water quality are *Deschampsia flexuosa* L., *Betula pubescens* Ehrh., *Deschampsia caespitosa* L., and Dicranum majus. Species associated with streams of medium water quality were *Ribes uva-crispa* L., *Geranium sylvaticum* L., *Salix alba* L., *Chelidonium majus* L., Ribes spicatum E. Robson, *Ranunculus lanuginosus* L. and *Viola mirabilis* L. High–medium-quality site-specific species were *Picea abies* L., *Hylocomium splendens*, and *Viburnum opulus* L. Species that significantly belong to streams of low quality were *Calamagrostis arundinacea* L., Galeopsis bifida Boenn., *Dactylis glomerata* L. and *Stellaria media* L. 

## 4. Discussion

Species composition in riparian forests can be linked to adjacent stream quality [23]. We studied species composition in riparian forests located next to the small streams to understand factors that affect adjacent stream water quality. Riparian forest ecosystem services provided by plants differ between seasons—during the vegetation season (late spring and summer), understory plant species take part in nitrogen uptake, but during decomposition and litterfall they provide nutrient leaching [22]. We collected vegetation data during July and August, while most of the understory species are present. This means that our results represent the situation during the vegetation season. Bryophytes are an important understory layer component in riparian forests, especially during the early spring and late autumn, and there is no vegetation season for vascular plants [38,39]. In northern latitudes during the autumn and spring seasons, mosses take part in nitrogen fixation more than in summer [40]. Previous studies have found that in northern conditions, *Sphagnum* spp. mosses are hosts for bacterial groups that may contribute to nitrogen fixation [52,53]. Our results show that oligotrophic conditions are preferable for *Sphagnum* spp. species and the total coverage was higher as well. The only species in our bioindicator analysis associated with high–medium water quality was *Hylocomnium splendens*. In northern latitudes, N fixation by mosses (cyanobacteria) depends on a fertility gradient, with the best fixation rates associated with low fertility [40]. In boreal forests, mosses dominate in understory the layer, regulating above- and below-ground temperatures and moisture [39,54]. Our data suggest that higher moss coverage was found in oligotroph soil conditions. *Hylocomnium splendens*, *Sphagnum* spp. *Dicranum* spp. and *Ptilium crista-castrensis* were found in the presence of *Picea abies* L. and *Betula pubescens* Ehrh. 

Previous studies have suggested that biodiversity might be important in the improvement of ecosystem services in riparian forests, but also that it is more essential to focus on the diversity and specifics of functional attributes than on species diversity per se [55,56]. In our sample plots we detected high species diversity, but when focusing on nitrogen-demanding species (Ellenberg value for N > 7), *Aegopodium podagraria* L., *Galeobdolon luteum* Huds., *Lycopus europaeus* L. and *Anthriscus sylvestris* L., they were found in the presence of *Alnus incana* L. and *Padus avium* Mill., which reduces nutrient leaching only during the vegetation season. Additionally, several studies have reported that alder stands provide nutrient leaching [57,58], even in riparian zones [57]. In this case, forests dominated by *Alnus incana* L. can lead to water over-enrichment (eutrophication) in streams, causing growth and reproductive limitations for invertebrates as well as other groups of species, while changing the stream (and down-stream) environment over time. Stream water quality is affected also by other factors, for instance, land management in non-forested areas (agricultural fields and urban areas). 

Bioindicator analysis indicated that *Alnus incana* L. was associated with low water quality as well as *Calamagrostis arundinacea* L., but conversely, *C. arundinacea* L. has the ability to reduce the excess nitrogen in soil [57,58,59]. Our data showed that *Calamagrostis arundinacea* L., *Deschampsia flexuosa* L. and *Deschampsia caespitosa* L. dominated in the presence of *Betula pubescens* Ehrh., *Picea abies* L. In addition, *Deschampsia flexuosa* L. and *Deschampsia caespitosa* L. were associated with high water quality in bioindicator analysis, but these species are sensitive to high nutrient concentrations. In comparison with ericaceous dwarf shrubs, *D. flexuosa* has a higher growth rate and greater maximum foliage height under high N availability in soil [60]. These grass species, in comparison with other recorded herbaceous species in our study, have a wider root system, providing higher soil mechanical stability and protecting soil from landslides. 

Total yield and stand density had a significant impact on species composition of herbaceous plants and bryophytes. Canopy structure in riparian forests regulates light availability below the crowns (affecting vegetation) as well as in streams. High light availability often leads to substantial and rapid increases in primary production of the stream [61,62]. However, a mosaic of light and shade conditions is required to increase the availability of micro-habitats for the diversity of water-ecosystem species [55]. Additionally, a limited amount of deadwood in streams serves the same purpose—providing micro-habitats. However, an increased amount of it causes sedimentation and, linked to increased nutrient leaching, leads to the slowing down of water flow, eutrophication, reduced oxygen concentration and environments unsuitable for numerous protected (umbrella) stream species such as brown trout and pearl mussel [16,17,18]. The application of forest management could lead to better conditions for aquatic organisms. For instance, stream macroinvertebrate density can be increased by selective logging by mimicking natural disturbances with an important aspect—no excess deadwood in streams [63]. High stand density, as demonstrated in our case study, affects understory vegetation and, if not managed, will lead to high deadwood input in streams at some point of the stand’s natural development [8,24,63]. Additionally, it may lead to lower individual tree stability, which creates a higher probability of wind damage, causing the same end result. Therefore, especially in sites with fertile soil and thus higher risk of nitrogen leaching, sparser stands with favourable conditions for species capable to reduce the excess nitrogen in the soil is preferable and can be created via targeted forest management. In poorer soils, multi-annual species, reducing runoff waters, need to be favoured. 

To provide adjacent streams with shade and avoid high water temperatures during the summer, broadleaved species are more suitable in comparison to conifers because of their wider tree canopy [64]. In addition, light availability can be regulated by tree canopy. Additionally, broadleaved species provide higher-quality leaf litter, improving stream conditions and energetic support to aquatic food webs [24], whereas the quality of needle litter is lower [8]. Broadleaved species also may reduce inorganic nitrogen leaching into the streams, but it is a species-specific case [1]. In our sample plots where conifers dominated, the solution could be to introduce birch trees. Additionally, bioindicator analysis showed that *Betula pubescens* Ehrh. was associated with high water quality. Furthermore, species and stand-age diversity can improve stream function dynamics [63]. Selective logging in oligotroph soils could change species composition in riparian forests leading to improvements in structural and species diversity [7,63]. In single-species stands dominated by broadleaved trees, the introduction of fast-growing species could be beneficial [63], for example, willows (*Salix* spp.), aspen (*Populus* spp.), rowan (*Sorbus* spp.) and cherry (*Prunus* spp.). On the other hand, the expansion of species adversely affecting water ecosystems should be controlled [63]. 

## 5. Conclusions

Riparian forests and streams are complex ecosystems supplementing each other. Specific species compositions of mosses, herbaceous plants, shrubs and tree species in riparian forests can provide suitable conditions for organisms in stream ecosystems regarding retention of nutrients and organic matter, energy and food resources and water temperature regulation. Nitrogen-demanding species (*Aegopodium podagraria* L., *Galeobdolon luteum* Huds., *Lycopus europaeus* L. and *Anthriscus sylvestris* L.) that were found in presence of *Alnus* spp. are mostly annual species providing nutrient uptake only during the vegetation season. Based on these results, in sample plots dominated by *Alnus* spp., the presence of bryophytes was low. A potential solution could be a limited introduction of *Salix* spp. (because *Salix* spp. are preferred by beavers), as this taxon has been reported as suitable for phytoremediation purposes. Introduction of *Betula pubescens* Ehrh. in coniferous stands could help to enhance shade in streams and improve energetic support to aquatic food webs by higher quality of leaf litter in comparison to conifers. Total yield and density had significant impact on species composition in studied sample plots, but more research has to be carried out to draw strict conclusions about suitable forest management perspectives that lead to the improvement of water quality. 

## Figures and Tables

**Figure 1 plants-11-01316-f001:**
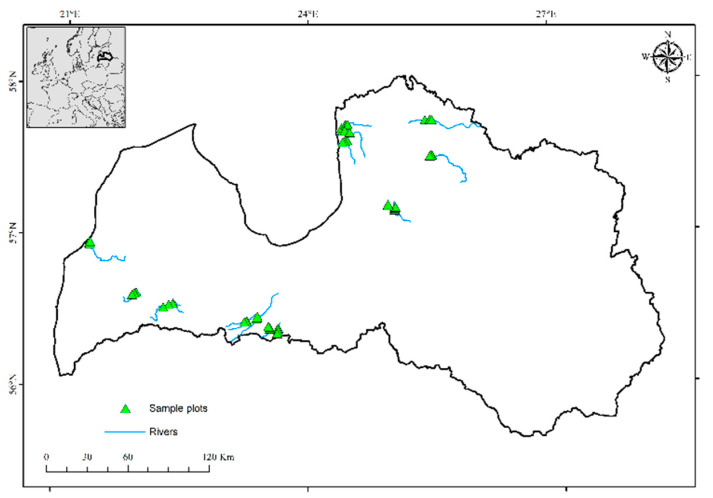
Location of selected sample plots and streams.

**Figure 2 plants-11-01316-f002:**
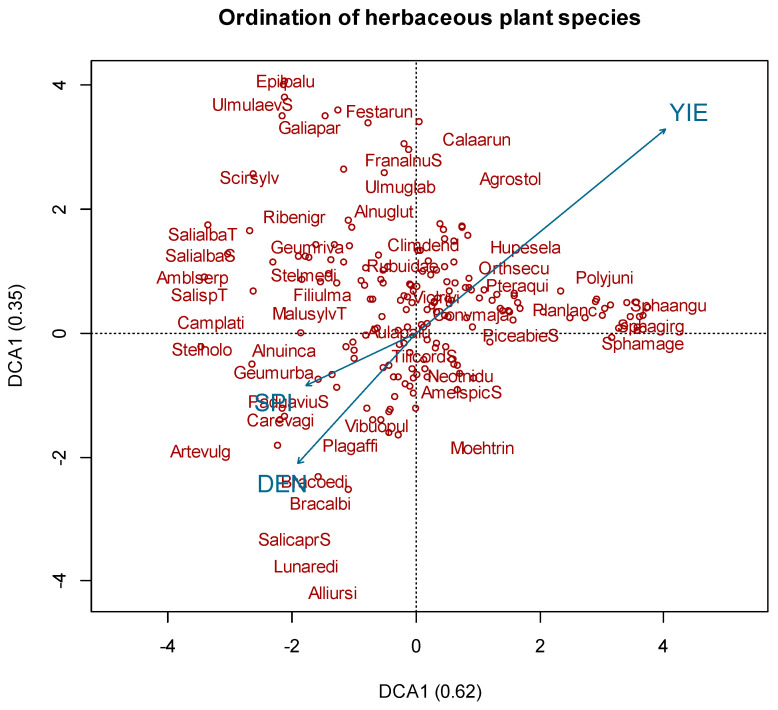
Grouping of understory vegetation species: results of detrended correspondence analysis. Analyzed factors: density (DEN), stream power index (SPI), total yeld (YIE). Letters “T” and “S” at the end of words represent the tree or shrub layer that they were classified into.

**Figure 3 plants-11-01316-f003:**
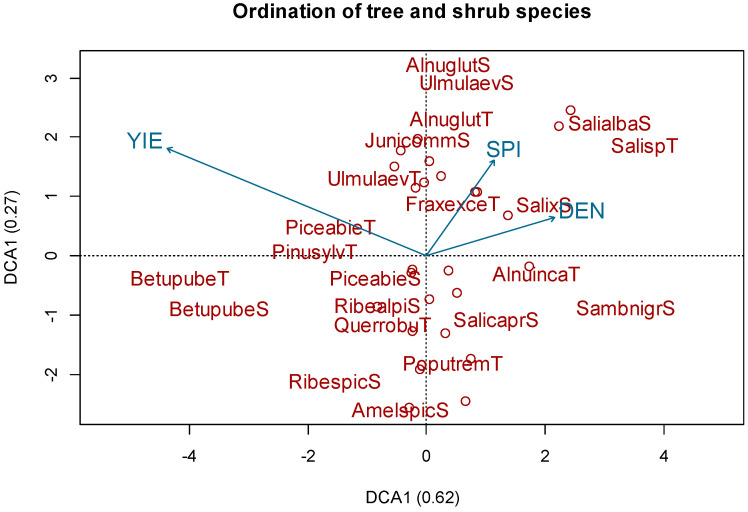
Grouping of tree and shrub species: results of detrended correspondence analysis. Analyzed factors: density (DEN), stream power index (SPI), total yeld (YIE). Letters “T” and “S” at the end of words represent the tree or shrub layer that they were classified into.

**Figure 4 plants-11-01316-f004:**
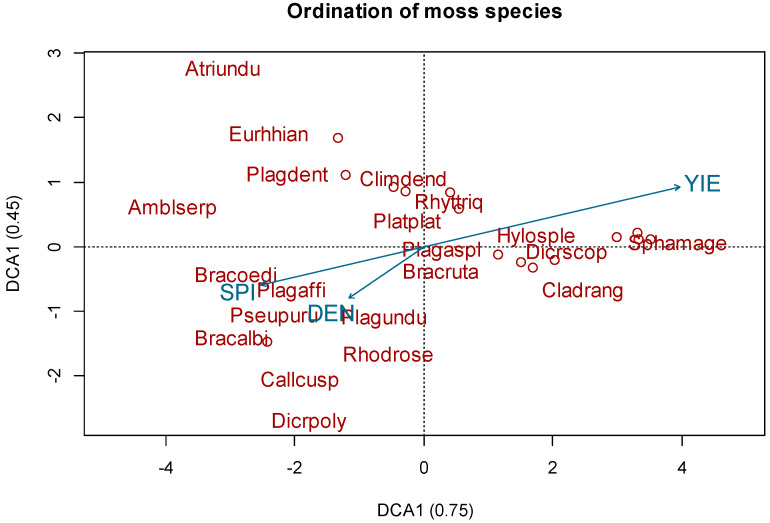
Grouping of moss species: results of detrended correspondence analysis. Analyzed factors: density (DEN), stream power index (SPI), total yeld (YIE).

**Table 1 plants-11-01316-t001:** ADONIS analysis—effects of total yield, total stand density, amount of deadwood and stream power index to understory species composition (including moss and plant species).

Factor	Df	SumOfSqs	R²	F Value	*p*-Value
**Total yield**	1	3.22	0.12	11.93	**0.001**
**Total density**	1	1.35	0.05	5.00	**0.001**
**Deadwood**	1	0.29	0.01	1.08	0.344
**Stream power index**	1	0.25	0.01	0.94	0.459
**Residuals**	84	22.69	0.82		
**Total**	88	27.81	1.00		

**Table 2 plants-11-01316-t002:** ANOSIM—changes in understory species composition related to the site (stream), distance from the margins, and water quality group.

Factor	R Value	*p*-Value
**Stream**	0.520	**0.001**
**Distance from the stream margins**	0.002	0.089
**Water quality**	0.150	**0.001**

## Data Availability

Not applicable.

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
