# Peer review of "Vegetation Affecting Water Quality in Small Streams: Case Study in Hemiboreal Forests, Latvia"

_plants, 2022, doi:10.3390/plants11101316_

Round 1

Reviewer 1 Report

L97 - not water quality groups but classes

L102/103 (and other) - forest massifs??? I guess You meant rather "forest complexes"

L113/114 too many surveys - rephrase these sentences

L117/118 - this info has already been given above

L130 rather "into one out of the following" as given stream probably was in one quality class at time

L170 in ANOSIM AN stands for analysis, so no need to repeat that word

L188 is R teh measure of variance?

L239-240 there is no analysis of the influence of stand parameters on stream water quality

figure 1 - replace "rivers" with "streams" to be consistent with the text

Author Response

Dear Reviewer,

We appreciate you for your precious time in reviewing our paper and providing valuable comments. It was your valuable and insightful comments that led to possible improvements in the current version. The authors have carefully considered the comments and tried our best to address every one of them.  Below we provide the point-by-point responses. All modifications in the manuscript have been marked using the “Track Changes” function.

Response to Reviewer 1

 [Comment 1] L97 - not water quality groups but classes

Response: Thank you very much for the comment. Revised accordingly.

[ Comment 2] L102/103 (and other) - forest massifs??? I guess You meant rather "forest complexes"

Response: Thank you very much for the comment. We agree that “forest complexes” is more appropriate term.

[Comment 3] L113/114 too many surveys - rephrase these sentences.

Response: Thank you very much for the comment. Revised accordingly.

[Comment 4] L117/118 - this info has already been given above.

Response. Thank you very much for the comment. Revised accordingly.

[Comment 5] L130 rather "into one out of the following" as given stream probably was in one quality class at time.

Response. Thank you very much for the comment. Revised accordingly.

[Comment 6] L170 in ANOSIM AN stands for analysis, so no need to repeat that word.

Response: Thank you very much for your nice reminder. Revised accordingly.

[Comment 7] L188 is R teh measure of variance?

Response: Thank you for the question! It is variance.

[Comment 8] L239-240 there is no analysis of the influence of stand parameters on stream water quality

Response: Thank you very much for the comment. We changed the sentence.

[Comment 9] figure 1 - replace "rivers" with "streams" to be consistent with the text

Response: Thank you very much for the comment. Revised accordingly.

Sincerely,

Stefanija Dubra,

stefanijadubra@gmail.com ,

Latvian State Forest Research Institute Silava, Rigas Street 111, LV–2169 Salaspils, Latvia 

Reviewer 2 Report

The introduction is unnecessarily long, but it can be accepted in this way.

Author Response

Dear Rewiewer, 

We appreciate you for your precious time in reviewing our paper. Thank you very much for the comment. We prefer to keep introduction as it is in current version.

Sincerely,

Stefanija Dubra,

stefanijadubra@gmail.com ,

Latvian State Forest Research Institute Silava, Rigas Street 111, LV–2169 Salaspils, Latvia

Reviewer 3 Report

-

Author Response

Dear Rewiewer,

We appreciate you for your precious time in reviewing our paper!

Sincerely,

Stefanija Dubra,

stefanijadubra@gmail.com ,

Latvian State Forest Research Institute Silava, Rigas Street 111, LV–2169 Salaspils, Latvia

This manuscript is a resubmission of an earlier submission. The following is a list of the peer review reports and author responses from that submission.

Round 1

Reviewer 1 Report

The study has low novelty. The section Material and methods is not sufficient. There is no properly described design of plots (transects) selection. Stream water quality should be described in detail, also. How was calculated stream power index? The water quality depends on situation of whole catchment. The plots are placed in different catchments, are individual catchments comparable? 

The aim of the work (to establish forest management recommendations promoting water quality in adjacent streams) is not supported by own data (the plots were established in stands without any signs of forest management in last decade). 

I am not native speaker, but the English needs improvement.

Reviewer 2 Report

Manuscript is focused on interesting and important topic. But current status needs revision.

Firstly, order of chapters is incorrect. Introduction (without title of chapter?), Material and Methods (it belongs to line 99 with number 2.), Results (number 3. is correct), Discussion (number 4. is correct), Conclusion (this chapter missing in the text). Also number for figures need revision (both have number 1.). For figures, links in the text missing.

Introduction is relatively long, but clear information, why this problem needs to be solved, missing.

Figures should be more readable (small letters, unclear map). It can be more described in subtitle of figures.

Supplement 1 is mentioned in line 139, but it is not included into manuscript (add it).

Sampling in two dates (July and August) in one year only seems to by limited (insufficient) in connection with study aims (compare also with line 160-162).

Line 203 – Which species are meant? Herbs, woody plants? If tree species are mentioned, the result can be opposite – species composition had impact on total yield….

Why species spruce and pine (line 247) are without full names and Latin names?

Why do the authors conclude that “floodplain forest management could help improve habitat diversity" (line 240-241), when their research was done "avoiding forest stands with any signs 252 of management activities during last decade" (line 252-253)?

Reviewer 3 Report

Well, the paper seems to be ok, but I miss sth here. The design is proper, but discussion is, although not indicated clearly, just a speculation. No research was done towards the various management issues, so you cannot draw strict conclusions. Moreover forest does not work in a single-factor mode: deadwood is not only a threat to stream organisms - on the other hand it is a "home" for many other species - we should consider here a balance and all the trade-offs

results are described only technically - you provide which analysis gives what, but you forget about issues that are behind the numbers

you mention supplement 1 - there is no supplement attached

L143 & 146 you mention components A and B, but nowhere they are explained

SPI variable is introduced, but what it is??